# Methods of Removal of Hormones in Wastewater

Daniela Guerrero-Gualan, Eduardo Valdez-Castillo [ID], Tania Crisanto-Perrazo *[ID] and Theofilos Toulkeridis *[ID]

Department of Earth Sciences and Construction, Universidad de las Fuerzas Armadas ESPE,
Av. General Rumiñahui S/N y Ambato, Sangolquí 171103, Ecuador
* Correspondence: ttcrisanto@espe.edu.ec (T.C.-P.); ttoulkeridis@espe.edu.ec (T.T.)

**Abstract:** Hormones are a type of emerging contaminant that reach the aquatic environment through wastewater effluents and which wastewater treatment plants (WWTP) cannot eliminate. The objective of this article was to determine the best hormone abatement technique between algae and microalgae, rotating biological discs, organic adsorbents, and activated carbon. For this, a critical review of the behavior of the abatement methods was conducted in the existing bibliographical scientific databases over the last eight years. Then, the Modified Saaty method was applied, establishing a relationship between removal efficiency, removal time, maintenance costs, stage of development, and environmental impact in each technique studied by a panel of experts, who weighted the chosen variables on a scale of 1–9 according to the variable's importance. The results indicated that the best technique to abate hormones is one that uses organic adsorbents and which reached a final comparative value of 0.58/1, which indicates the suitability of the method to combine the five comparison variables. At the same time, the rotating biological disc technique reached a value of 0.17/1, indicating its deficiency in the balance between the analyzed variables.

**Keywords:** hormones; wastewater; hormone removal; Saaty

## 1. Introduction

In recent years, water pollution has become a global problem, with increasing levels having negative effects on a variety of ecosystems [1–5], and affecting the survival of aquatic species and human health [6–11]. Hormones, which are a type of emerging pollutant (EC), are considered to be persistent substances in the environment [12–23]. They have probably been present in environmental matrices for as long as human beings have been using them [24]. These contaminants are endocrine disruptors (EDC), that is, they repeat the behavior of endogenous hormones and cause alterations in the endocrine system of living organisms [25–30]. The contamination of aqueous media by hormones has become a growing concern in different regions of the planet, raising interest in new methodologies for their mitigation and removal since traditional treatments in waste waters (WW) do not seem to be sufficient [31]. This contamination has been shown in the negative effects experienced by humans and wildlife on their endocrine system [32].

Hormones are classified into three groups: female (estrogens), male (androgens), and gestational (progestogens) [33–35]. However, the hormones that are most present in wastewater (WW) are estrogens, which can be natural (estrone: E1, and 17β-estradiol: E2) or synthetic (17-α-ethinyl estradiol or EE2) [36–39]. The estrogen molecule is based on the structure of phenanthrene which is produced in the ovaries of female vertebrate and invertebrate animals [40]. The bioconcentration and biotransformation of EC in WW has caused them to be of great concern to scientific communities [41–43]. The exposure of human beings to these substances primarily occurs because of the high concentrations of estrogens in drugs used for hormone replacement as well as contraceptive methods or regulators of the menstrual cycle. These hormones are produced in the body of animals and humans; in the sexual organs, they are produced by the testes, placenta, and ovaries [44,45].

The discharge of estrogens in WW is 30,000 kg/year [33]. These include synthetics consumed by humans, especially in contraceptive pills, however, the contribution of cattle is much higher [33,46,47]. Purification plants or wastewater treatment plants (WWTP) are not designed for the removal or control of these compounds and therefore do not have the necessary efficiency to eliminate them; therefore, the substances reach the environment through WW without any treatment [48–50]. This is because the elimination of this contaminant in the WWTP is incomplete and its effluent becomes a source of chemical contamination despite its concentrations being in the range of ng/L and causing biological effects [48,51,52].

Recent research indicates that the treatment of these substances in WWTPs is not possible because of the toxicity repudiation and because the metabolites have become stronger over time [19,53]. For example, using conventional techniques for WW treatment as an advanced oxidation process could form even more harmful by-products than the original compounds [54,55]. Since these procedures only partially remove EDCs, they are continuously introduced to the aquatic environment [14] where they cause problems with metabolism, change normal reproduction, and interfere with homeostatic control in some animals [56]. For human beings, this is a public health issue because WW is discharged into drinking water sources in several places (de facto reuse) [57].

Hormones can be reduced by WWTP with physical, biological, and advanced oxidation methods [58–60]. In the removal of steroid hormones, biodegradation is considered to be the main mechanism, which is highly effective with abatement rates of between 91% and 100% in hormones such as androgens and progestogens. In estrogens, the efficiency is lower (67–80%) [53]. Recent research indicates that this type of contaminant enters the environment through WW. However, many treatment plants or WWTP do not have the necessary efficiency to eliminate these substances. It is therefore necessary to implement highly effective techniques in the removal of hormones in WW [48,51,52].

Other methods, such as oxidation, ultrafiltration, and nanofiltration are neither technically nor economically viable because they are difficult to execute, have complex processes and maintenance, and have high investment costs [58].

The advanced oxidation process (AOP) includes processes such as ozonation, Fenton, photo-Fenton, and others, such as radiation and ultrasound, which depend on the generation of OH [59]. This technology is considered one of the most promising for the removal of hormones in WW compared to other conventional biological treatment techniques [60]. The main drawback is the generation of toxic by-products and the high energy consumption during the membrane separation process. Therefore, these techniques have a negative impact on the environment [61].

At the same time, ultrafiltration systems are technologies that operate at relatively low pressures but cannot remove contaminants with subnanometric sizes, such as steroid hormones. Therefore, this technique is usually hybrid, that is, it is usually accompanied by other processes, such as nanofiltration, to achieve removal ranges between 50% and 75% [60]. Currently, this technique is in the bench scale phase [62].

Based on the need to use novel techniques in hormone removal [63], the methods discussed in the current study are still limited to laboratory and pilot phases. Nonetheless, combined methods are being developed (biological processes and membrane separation), such as membrane bioreactors that have been implemented on a large scale and which allow the elimination of hormones, such as estrogen [64,65].

The analyze the choice for the most beneficial method to abate hormones in WW, the parameters should be evaluated with decision tools that use fuzzy analytical hierarchy or Modified Saaty (FAHP) methodologies [66], which minimize the uncertainties that occur within group decisions [67–71]. These analytical hierarchy processes (AHP) or Saaty methodology allow decisions to be made using multiple criteria that are based on the comparison by pairs and alternatives with the discussion of a panel of experts in the characterization of the topic, facilitating the evaluation of priorities [67,72,73]. This technique

has allowed for decisions to be made regarding improvements in environmental issues as well as incorporating its criteria in many academic fields, especially engineering [74].

Based on this, the main objective of this work has been to compile the state of the art in different methods of hormone abatement in WW using a comparison with FAHP. These include the parameters of efficiency, maintenance cost, removal time, laboratory stage, and environmental impact of the WW treatment techniques of a WWTP.

## 2. Materials and Methods

Bibliographic reviews of the SpringerLink and Science Direct scientific databases were used for the corresponding investigation, performing a critical review of the literature in which the information of the last two decades of the state of the art of hormone abatement and the technologies was extracted for the remediation of waters that present this type of EC. Subsequently, a comparison of efficiency, removal time, maintenance cost, laboratory stage, and environmental impact was conducted, using the modified Saaty method, and the most convenient technique for hormone abatement was obtained as a result.

### 2.1. Remediation or Removal

#### 2.1.1. Biological Processes

Since they are robust, cost-effective, and have a low environmental impact, biological processes are considered the most common treatment for WW [75–78]. This type of process takes advantage of the ability of microorganisms to use wastewater components to provide energy for their metabolism and thus eliminate contaminants [79].

#### Algae and Microalgae

The use of algae and microalgae are considered a new process for treating WW, and which is in the laboratory phase [80–83]. Microalgae have effective biological systems that convert solar energy into organic compounds (organic matter), and have the ability to remove micropollutants through bioadsorption, bioaccumulation, biodegradation, photodegradation, and cometabolism [84]. From studies in countries such as Australia, microalgae are considered to be organisms with a great capacity to eradicate contaminants through various mechanisms that have resulted in reduced impacts through the recovery of bioproducts. However, this is not a very efficient technique for removal of hormones in WW [81,85].

The algae and microalgae method is effective due to the reduction of hormones present, and the low cost of its implementation and energy consumption, in addition to the low environmental impact. By using this method, the production of useful compounds for the environment is high [79]. The depletion mechanism is due to the capacity of these microorganisms for rapid initial absorption, bioaccumulation, and biodegradation; it eliminates EDCs with more than 90% efficiency in E2 and EE2 within the first 12 h [86]. Figure 1 illustrates the depletion of hormones in WW.

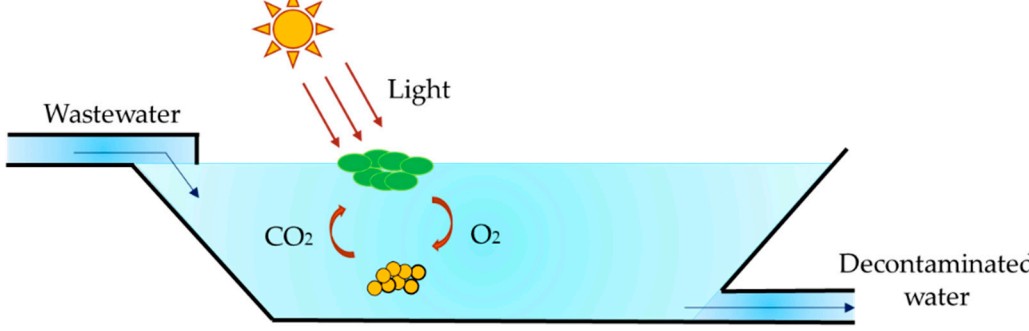

**Figure 1.** Hormone suppression through the use of algae and microalgae [87].

To obtain the behavior of the method, Lu et al. (2021) [79], used algae (*C. lentilifera, U. pertusa, G. lemaneiformis*, and *C. fragile*) incubated at 20 °C with dark cycles which was then passed through fiberglass filters before being used. Abiotic loss of E2 or EE2 was evaluated by abiotic control assay with only E2 or EE2. Additionally, it was necessary to evaluate other variables, including salinity, photoperiod, temperature, and nutrient concentration [81]. The total time of the procedure was 24 h when using only one cycle, however, all experiments were performed in triplicate [86].

Rotating Biodisc Contactor (RBC)

The rotating biodisc contactor (RBC) is a treatment capable of eliminating high concentrations of specific hormones such as E2 and EE2; it has an average abatement efficiency of 66% [88–92]. RBC is a biological method in which the rotation of the disk provides the necessary support for the creation of a consistent biofilm, guaranteeing the oxygenation of the system [86,92].

In many of the cases in WWTPs, these rotating discs are manufactured to be environmentally friendly, which is why there are few load losses [86]. The use of RBC is considered a biological treatment; its application is complex but influential in removing not only hormones but also other soluble contaminants in WW [93,94].

The pilot scale method has several advantages for its use, including the low cost of implementation, low maintenance, and low energy use [88]. The area required for assembly is also small [95]. To observe the efficiency of the method, Maurício et al. (2018) [86] divided the study into three experimental phases. In phase 1, the biodegradation of each compound was examined. In phase 2, the same process was studied, but this time as a mixture. In phase 3, to study the degradation of the WWTP water, the development was executed in a different matrix with the analysis time of about 10 min. The depletion time of EE2 was 6 h, achieving a maximum efficiency of 68%; for E2, the maximum removal occurred at 3 h and a percentage of 46% [86]. The laboratory stage consisted of two RBC systems, with four Teflon discs in which the homogeneity of the conditions and the complete aeration of the content were guaranteed [88].

2.1.2. Physical Processes

Organic Adsorbents

Adsorption techniques with organic components are attractive in environments where there is agricultural production because these by-products do not require numerous steps for their processing [96]. This treatment is suitable for WW because it is cheap, versatile, and simple. It does not consume high amounts of energy, and it allows for better quality effluents [97]. Figure 2 illustrates the operation of this system.

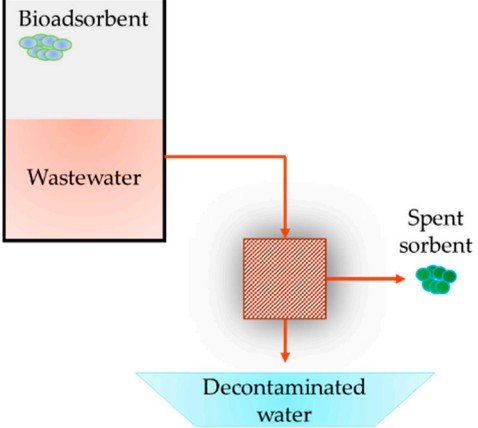

**Figure 2.** Functioning of the hormone abatement technique through organic adsorbents [96].

The system is efficient in the abatement of hormones since it presents satisfactory adsorption capacities of over 80% [96]. Honorio et al., (2018) [95] proposed using rice hulls and soybean hulls for the extraction of natural E1, E2, and E3 that are contained in the effluents of the pig industry. To analyze these contaminants, specific samples were taken from four effluent points that were characterized in the following order: (1) tank with a sieve that collects the coarse material that is taken to a dryer, (2) heaps of manure with a capacity for 1 day of effluent, (3) a sequence of four lagoons, and (4) a wetland. In the same way, samples of the hormones were taken through the solid-phase extraction (SPE) method [95].

In the abatement stage, a mixture of rice hulls and another of soybean hulls was used. The adsorption experiments were performed in duplicate with 0.1 g of biomass in natura in contact with 25 mL of the multicomponent solution, without adjusting the pH, while the contact time was of about 4 h. Therefore, the samples were filtered with cellulose ester membranes (0.45 μm) and quantified without performing the extraction procedure [96]. By using organic waste, this process has a low environmental impact.

Powdered Activated Carbon (PAC) and Granular Activated Carbon (GAC)

Activated carbons are compounds that have an extended internal surface and a high degree of porosity. These characteristics have allowed their application in the removal of organic matter in WW, from high to very low concentrations [61]. The method that is observed in Figure 3 is in the laboratory phase and is considered to have one of the highest removal capacities because of the simplicity of its design, the reduced use of energy, and low investment cost; its environmental impact is also noted since it does not generate toxic by-products [98].

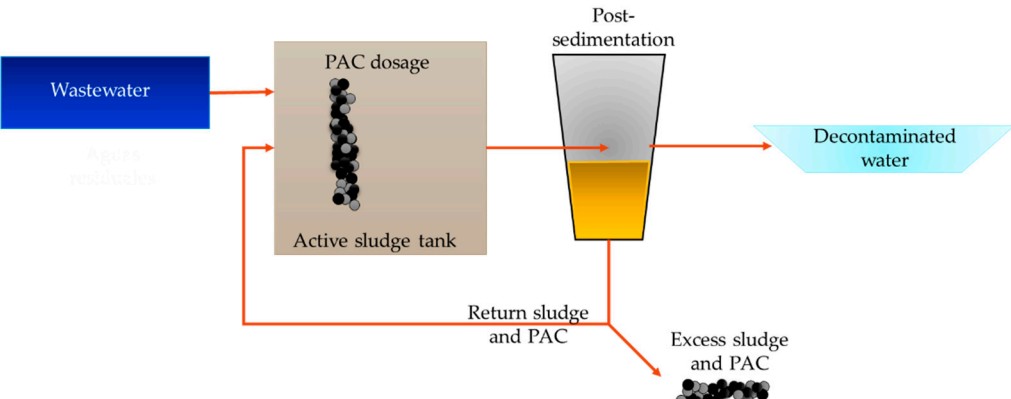

**Figure 3.** Knockdown operation with PAC and GAC [99].

Esmaeeli (2017) [55] formulated the elimination of estradiol valerate (EV) and progesterone (PRO) from aqueous solutions, starting by evaluating the effects of the initial pH solutions, the amount of adsorbent, the contact time, the initial concentration of EV and PRO and temperature. The pseudo-first order and pseudo-second order kinetic models fit the experimental data precisely. The highest level of adsorption percentage of contaminants occurs within the first 480 min. The maximum adsorption capacity and thermodynamic parameters showed that GAC and PAC can be selected as appropriate and effective pharmaceutical WW treatment techniques [63]. This technique has been used for the abatement of the hormones EV and PRO with an efficiency of 98% removal [99]. In the same research, Esmaeeli et al., (2017) [55] used scanning electron microscopy (SEM) and Fourier transformed infrared spectroscopy (FTIR) as complementary techniques to detect changes that are caused by adsorption.

PAC and GAC techniques have recently been coupled with ultrafiltration processes (UF) in order to improve systems and allow for the removal of contaminants in WW. The PAC is responsible for guaranteeing the quality of the effluent and the GAC to mitigate the

fouling by UF and the deposition of PAC [55,100,101]. Another of the biological treatments that this technique puts in place is the activated sludge process commonly implemented in WWTPs; the level of abatement efficiency of this type of micropollutants has caused them to be replaced by these new techniques [102].

### 2.2. AHP and FAHP

For the application of the FAHP technique [61], a panel of experts chose the variables and assigned values from higher to lower importance, being maintenance cost 9, removal time 7, efficiency 5, environmental impact 3, and stage of development 1. The weights were made using a cross matrix, considering each of the variables of interest and the rating given by the panel of experts. Thus, for example, to determine the weight or ponderation of the RE (removal efficiency) variable, an analysis is conducted against the other parameters and their respective score; considering the RE versus the RE its rating would be 1/1, which is equivalent to 1; the RE compared to the MC (maintenance cost) would be 5/9, equivalent to 0.56, compared to the SD (Stage of development), 5/1 equal to 5, compared to the EI (Environmental impact), 5/3, and comparing with the RT (Removal time), it would be 5/7, which is equivalent to 0.71. Once the crossed matrix of the parameters under consideration is completed, for each row (variable), what is known as power is found and determined with Equation (1), where n corresponds to the number of variables to be considered, for the case under study five.

$$Ci = (RE \times MC \times SD \times EI \times RT) \wedge (1/n) \tag{1}$$

where $Ci$: weight factor per variable, RE: removal efficiency, MC: maintenance cost, SD: stage of development, EI: environmental impact, RT: removal time, and n: number of variables. In order to determine $Ci$, the product of each of the variables is considered, and the analysis by its degree of importance is raised to the inverse of the number of variables contemplated.

To obtain the weight $w_i$, a sum of all the values obtained in the parameters is made, dividing the partial weight for the sum; in other words, for the case of the RE variable, it was divided $1.27/6.35 = 0.20$. The methodology suggests calculating the level of consistency of the calculations made, since it is necessary to verify that the matrix does not have contradictions between the criteria assigned to each variable [103]. To realize this, the weight $w_i$ of the analyzed variable must be multiplied by the sum of the weighting to obtain the final sum, which must be the equivalent of the number of variables used in the matrix, as indicated in Equation (5).

$$\lambda i = w_i \times p_j \tag{2}$$

where $\lambda i$: consistency level, $w_i$: weight and $p_j$: sum of the weighting of each variable

Thus, for example, for the case of RE, $\lambda i$ is equal to $0.20 \times 5 = 1$. The development of the analysis in each of the variables is explained in Table 1.

**Table 1.** Weighting of Criteria.

| Criteria | RE | MC | SD | EI | RT | $C_i$ | $w_i$ | $\lambda_i$ |
|----------|------|------|-------|------|------|------|------|------|
| RE | 1.00 | 0.56 | 5.00 | 1.67 | 0.71 | 1.27 | 0.20 | 1.00 |
| MC | 1.80 | 1.00 | 9.00 | 3.00 | 1.29 | 2.29 | 0.36 | 1.00 |
| SD | 0.20 | 0.11 | 1.00 | 0.33 | 0.14 | 0.25 | 0.04 | 1.00 |
| EI | 0.60 | 0.33 | 3.00 | 1.00 | 0.43 | 0.76 | 0.12 | 1.00 |
| RT | 1.40 | 0.78 | 7.00 | 2.33 | 1.00 | 1.78 | 0.28 | 1.00 |
| Σ | 5.00 | 2.78 | 25.00 | 8.33 | 3.57 | 6.35 | 1.00 | 5.00 |

The level of consistency should be as close to one as possible, because the further away it is means that the criteria used for its weighting and/or normalization were not adequate,

or it may be an indication of the randomness of the criteria [104]. Once Table 2 has been developed, the weights of the comparison variables, Equation (3), is applied with the aim of evaluating each method according to the assigned criteria.

$$\text{Methodology} = w_1 \times W_1 + w_2 \times W_2 + w_3 \times W_3 + w_4 \times W_4 + w_5 \times W_5 \tag{3}$$

where $w_1$: weight of removal efficiency, $W_1$: removal efficiency, $w_2$: weight of maintenance cost, $W_2$: maintenance cost, $w_3$: weight of stage of development, $W_3$: stage of development, $w_4$: weight of environmental impact, $W_4$: environmental impact, $w_5$: weight of removal time and $W_5$: removal time.

**Table 2.** Analysis parameters of the studied techniques.

| | Removal Efficiency (%) | Removal Time (h) | Stage of Development | Maintenance Cost | Environmental Impact |
|---|---|---|---|---|---|
| Algae and microalgae | 90 | 12 | Laboratory | Low | Low |
| Rotating biodisc contactor | 66 | 4.33 | Pilot | Low | Low |
| Organic adsorbents | 80 | 4 | Pilot | Low | Low |
| Powdered activated carbon and granular activated carbon | 98 | 8 | Laboratory | Low | Medium |

Once the benefits of each of the hormone abatement techniques of interest were known, we proceeded to analyze the one that performs best within the studied criteria. Each criterion assumed is represented by measurement scales of different qualitative and quantitative value. Therefore, it is necessary to normalize these values with the use of the mathematical formulation represented in Equation (1), in which each element takes a membership value according to the degree of domain that it takes in relation to the total of the set under study [66]. Thus, the limits of each criterion indicate the minimum and maximum values of the factor, in which it is estimated that any intermediate value corresponds to a degree of probability [68].

$$N = (V_o - V_{min})/(V_{máx} - V_{mín}) \tag{4}$$

where N: variable normalization, $V_o$: original value, $V_{min}$: minimum value and $V_{máx}$: maximum value.

The normalization of the variables under study corresponded to the standardization in terms of probability, that is, the values were converted within a range of 0 to 1. For this, it is necessary to understand the behavior of the variables with what is to be obtained, which is the most adequate method of hormone removal. The use of Equations (5) and (6) depends on the direct or inverse influence between the analysis criteria vs. the study technique. Once the relationship is established, the sine or cosine functions [67] are applied.

$$\mu_A(V_0) = \sin(\pi/2 \times N) \tag{5}$$

$$\mu_A(V_0) = \cos(\pi/2 \times N) \tag{6}$$

where $\mu_A$: evaluation of the criterion, $V_0$: original value and N: variable normalization.

For example, for the calculation of the elimination efficiency parameter, the normalization was directly proportional, that is, the higher the elimination or removal efficiency (RE) of the method, the more convenient the method; therefore, the corresponding membership function is the sine curve, as illustrated in Figure 4.

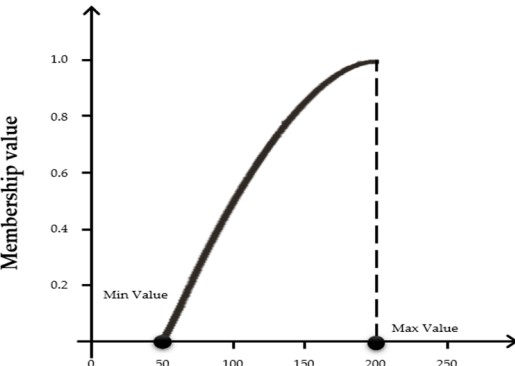

**Figure 4.** Membership function corresponding to the sine curve [61].

Nonetheless, the maintenance cost variable (MC) is calculated with the cosine function (Figure 5) because of its inversely proportional behavior, which means that the lower amount required for maintenance it is an advantage.

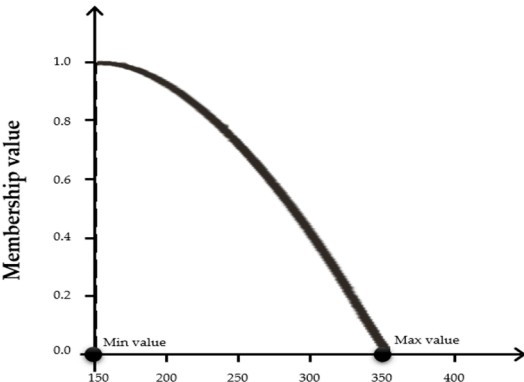

**Figure 5.** Membership function corresponding to the cosine curve [61].

## 3. Results

According to the literature review, the methods previously studied for the remediation of water contaminated with hormones have achieved effective and encouraging results. In the methodologies presented, it was observed that the efficiency for the elimination or degradation of hormones depends on several factors, including the biological and chemical persistence and the physicochemical properties of the target contaminant. Table 2 lists the characteristics of the four techniques analyzed for their comparison and determination of the best method.

Applying to the case study, Equation (7) is obtained.

$$\text{Methodology} = 0.20 \times W_1 + 0.36 \times W_2 + 0.04 \times W_3 + 0.12 \times W_4 + 0.28 \times W_5 \qquad (7)$$

where $W_1$: removal efficiency, $W_2$: maintenance cost, $W_3$: stage of development, $W_4$: environmental impact and $W_5$: removal time.

In order to obtain these values, the evaluation criteria or parameters are considered in each of the techniques under the same criteria evaluation mode. Each of these has different behavior because the direct or indirect behavior of the technique is examined to apply the functions corresponding to the sine or cosine (Equations (2) and (3)). For example, for the abatement that was due to the use of algae and microalgae, each of the evaluation criteria was analyzed; two of its variables, RE and SD, have a directly proportional behavior, while MC, EI and RT behave in an inversely proportional manner. Therefore, for hormone depletion in WW, the higher the RE, the more applicable the technique. On the other hand, when the MC is greater, the method is less applicable. According to the behavior of each of the evaluation criteria, the membership equation was applied; this considers

an evaluation degree corresponding to a minimum and maximum value. For example, for RD, the following evaluation grades were considered: poor, fair, good, and excellent, corresponding to 66–74.5%, 74.5–83%, 83.91–91.5% and 91.5–100%, respectively.

Saaty allows weighting this evaluation in odd values from 1 to 9 and generating its weight by using the concept of partial fraction; for the RE variable case, the results were 0.06 for poor, 0.17 for regular, 0.28 for good and 0.50 for excellent. Once the partial fractions were obtained, the sinusoidal or cosinusoidal behavior of the criterion was considered under each of the techniques. In the case of abatement with algae and microalgae, RE has a directly proportional behavior and was classified with a good criterion, that is, RE was between 83 and 91.5%. Therefore, Equation (2) was applied, where the minimum value was given by the weight that corresponds with bad, equivalent to 0.06, while the maximum value corresponds to 0.50 and the original value is 0.28. Once the minimum, maximum, and original values were identified, the membership function was applied, obtaining one of the elements for the weighted linear sum and considering what needs to be completed for each evaluation criterion. For the case of algae and microalgae, the value of $w_i$ in the RE is 0.707 and is multiplied with the weight obtained in Table 2. This value is 0.20 and, in this way, was the first addend of the technique equal to 0.14.

In a similar way to that discussed above, it is evident that, with each of the criteria considered, what differs is the degree of evaluation, and therefore its weighting. Thus, for MC, the low, medium, and high grades were considered with a weight of 0.60, 0.33, and 0.07 respectively. For regular SD, good and very good with weights of 0.07, 0.33, and 0.60; the low, medium, and high EI have a weight of 0.60, 0.33, and 0.07, respectively; the RT of poor, regular, good, and excellent had weights of 0.13, 0.21, 0.29, and 0.38, respectively. Through the application of Equations (2) and (3) as mentioned above, the values of each of the summands of the weighted linear sum were obtained; finally, the result is the sum of the terms in each of the criteria for the abatement that they contemplate. This procedure was repeated for each of the abatement techniques considered in this study. The results obtained for each method are summarized in Table 3.

**Table 3.** Weighted linear sum of the methods analyzed.

| Method = | $W_1 \times$ RE | + | $W_2 \times$ MC | + | $W_3 \times$ SD | + | $W_4 \times$ EI | + | $W_5 \times$ RT | = | RESULT |
|---|---|---|---|---|---|---|---|---|---|---|---|
| Algae and microalgae | 0.14 | + | 0.00 | + | 0.00 | + | 0.00 | + | 0.28 | = | 0.42 |
| RBC | 0.00 | + | 0.00 | + | 0.03 | + | 0.00 | + | 0.14 | = | **0.17** |
| Organic adsorbents | 0.08 | + | 0.36 | + | 0.03 | + | 0.12 | + | 0.00 | = | **0.58** |
| PAC and GAC | 0.20 | + | 0.00 | + | 0.00 | + | 0.12 | + | 0.24 | = | 0.56 |

## 4. Discussion

Table 3 lists the results obtained after applying the Modified Saaty technique. Regarding the ERE, the RBC method obtained the lowest weight, and PAC and GAC obtained the highest. In the maintenance cost (MC), the weights of the techniques were zero except for organic adsorbents, because when applying Equations (2) and (3) the result was 0. For SD, the higher weights are found both in RBC and in organic adsorbents. Regarding EI, the highest weights were for organic adsorbents and for PAC and GAC; for the removal time, the algae and microalgae method stands out, and has the lowest time for organic adsorbents.

In the bibliographic review, it was observed that the PAC and GAC method had a hormone removal efficiency of 98%, followed by the algae and microalgae techniques with 90%, organic adsorbents with 80%, and the use of RBC that removes hormones with an efficiency of 66%. Another of the variables considered was the time necessary for the hormone to be eliminated, so that in the analysis carried out, the method with the shortest removal time is organic adsorbents and RBC with 4 and 4.33 h, respectively, followed by the PAC and GAC methods with 8 h of removal, while with algae and microalgae, the contaminant abatement was 12 h. Regarding the variable stage of development, RBC techniques together with organic adsorbents are in a pilot plan, which gives greater weight to the final sum in terms of establishing the most convenient method. This has been contrary

to the use of algae and microalgae or PAC and GAC, which, being in the laboratory phase, decreases the convenience of using these methods. Unfortunately, none of the methods analyzed have been developed on a large scale.

Regarding the MC criterion, all the techniques contribute the same value in the final weight because they do not represent high costs with respect to their use. In the same way, the EI variable is related, since from the previously reviewed bibliography, the techniques throw few polluting loads into the environment, especially when organic adsorbents are used; this is the same with GAC and PAC. In the first instance, it would be very difficult to choose the most convenient method for hormone depletion in WW because the methods all have different determination values for the studied criteria. The use of FAHP provides a methodology to consistently determine the best technique for contaminant removal. Once the modified Saaty FAHP has been conducted, the criteria of removal efficiency, removal time, maintenance cost, stage of development, and environmental impact are considered to establish the most viable, least expensive, and most convenient system for the removal of hormones in WW.

In the current study, technique 3 (removal of the contaminant by organic adsorbents) obtained a rating of 0.58/1, making it the methodology that best or most adequately combines the analysis criteria, and the one that presents the most benefits in its use, that is, it is the most effective in contrast to the other methods being studied. This occurs even though the technique does not have the highest percentage of removal efficiency compared to the other techniques. However, the other criteria serve to qualify it as the most beneficial method even though the maintenance cost is the variable with the greatest weight, since it creates a lower cost; this implies a greater attraction for its implementation.

The PAC and GAC methods have a result of 0.56/1 because the variables maintenance cost and stage of development do not add any weight to the procedure. In the same way, for the removal of hormones in WW, the use of algae and microalgae is close to the most efficient process with a result of 0.42/1. However, it is worth emphasizing that the only variables considered for this are efficiency and removal time; based on these characteristics of the process and their behavior according to the used methodology, the other three variables were not included in the analysis.

On the other hand, the method that least complies with a balance between the variables analyzed is RBC with a score of 0.17/1. The variables of removal efficiency, maintenance cost, and environmental impact do not have a weight within the weighted linear sum of this technique. Similarly, the removal time is one of the lowest among the different techniques. The use of the FAHP methodology allows for optimal management of resources, choosing the appropriate techniques for the abatement of emerging pollutants, and avoiding unnecessary costs as well as reprocesses [105].

## 5. Conclusions

Currently, the use of hormones has become a topic of great interest for researchers because of the effects they cause both in humans and in animals. These contaminants behave as endocrine disruptors, and their concentrations and designs of the WWTPs mean that eliminating them from their effluents is impossible; this has led to the use of methods for their elimination. However, to choose an appropriate hormone removal technique, all criteria that are important to the user must be balanced.

The present study determined that the best method for the abatement of hormones is using organic adsorbents, since it reached a result of 0.58/1. This means that the analysis criteria keep an average balance with respect to the optimum of each criterion. Likewise, the method for the depletion of hormones that presents a low interrelationship between the optimum of the criteria analyzed is the RBC method, with a value of 0.17/1, so its implementation must be carefully reviewed prior to particular application.

It is highly recommended that, for the application of the Modified Saaty (FAHP) methodology, there is a panel of experts who understand the techniques and advises on

the qualitative–quantitative values that are assigned to the study variables and can thus achieve consistency closest to 1.

**Author Contributions:** Conceptualization, T.C.-P., D.G.-G. and E.V.-C.; methodology, T.C.-P.; software, D.G.-G. and E.V.-C.; validation, T.T. and T.C.-P.; formal analysis, T.C.-P., D.G.-G. and E.V.-C.; investigation, T.T. and T.C.-P.; resources, T.C.-P., D.G.-G., E.V.-C., and T.T. writing—original draft preparation D.G.-G. and E.V.-C.; writing—review and editing, T.T. and T.C.-P. visualization D.G.-G. and E.V.-C. All authors have read and agreed to the published version of the manuscript.

**Funding:** This research received no external funding.

**Institutional Review Board Statement:** The study did not require ethical approval.

**Informed Consent Statement:** Not applicable.

**Data Availability Statement:** Not applicable.

**Conflicts of Interest:** The authors declare no conflict of interest.

## Abbreviations

| | |
|---|---|
| AHP | Analytical hierarchy processes |
| AOP | Advanced oxidation process |
| Ci | Weight factor for variables |
| E1 | Estrone |
| E2 | 17β-estradiol |
| EC | Emerging pollutants |
| EDC | Endocrine disruptors |
| EE2 | 17-α-ethinyl estradiol |
| EI | Environmental impact |
| EV | Estradiol valerate |
| FAHP | Fuzzy analytical hierarchy or Modified Saaty |
| FTIR | Fourier transformed infrared spectroscopy |
| GAC | Granular activated carbon |
| li | Consistency level |
| MC | Maintenance cost |
| n | Number of variables |
| N | Variable normalization |
| PAC | Powdered activated carbon |
| pj | Sum of the weighting of each variable |
| PRO | Progesterone |
| RBC | Rotating biodisc contactor |
| RE | Removal efficiency |
| RT | Removal time |
| SD | Stage of development |
| SEM | Scanning electron microscopy |
| UF | Ultrafiltration processes |
| Vmáx | Maximum value |
| Vmin | Minimum value |
| Vo | Original value |
| wi | Weight |
| WW | Waste waters |
| WWTP | Wastewater treatment plants |
| μA | Evaluation of the criterion |

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
