# Peer review of "Methods of Removal of Hormones in Wastewater"

_water, doi:10.3390/w15020353_

Round 1
Reviewer 1 Report
General remarks
Manuscript discusses different methods of hormones removal from wastewater and evaluates their usefulness using multiple criteria analysis. In the manuscript clear information about techniques applied in the technical scale is missing. The text makes impression that all mentioned techniques are still in laboratory or pilot scale. It needs completing and the recent development of the techniques of micropollutants removal as well as their current application in the technical scale should be taken into account in the presented analysis.
Discussed four techniques should be compared with an efficiency of an activated sludge being the most popular method of wastewater treatment. Hormones' sorption on activated sludge should also be discussed if active carbon applied directly to biological reactor as a technique is considered.
Specific comments
Page 2, line 54 – „ignorance of toxicity” – this problem should be more deeply presented.
Lines 56-59 – this sentence presents opposite opinion to the sentence given in lines 46-49 and lines 53-54, where lack of efficient removal of hormones at WWTP is stated. WWTPs base on biological processes.
Lines 59-64 – deeper discussion of the application of mentioned techniques in the technical scale is necessary.
Lines 93-94 – Wastewater treatment with microalgae is also applied in the technical scale. It is popular for instance in Australia and US. Some facts about it should be added.
Figure 2 is not informative. It doesn’t present the RCB construction. If a reader doesn’t know this technology, this figure doesn’t help him to understand it. If better version of the RCB is not available it would be better to remove Figure 2 at all.
Lines 116/117 – the sentence needs rewriting. In Figure 2 any load losses aren't seen.
Line 127 – what kind of analysis the authors mean?
Lines 145/146 – What kind of both the effluents the authors mean?
Author Response
Response to Reviewer #1
Dear Reviewer,
As authors of the manuscript “Methods of Removal of Hormones in Wastewater”, we appreciated a lot your suggestions and comments on the document, as we are certain and convinced, that they have been useful to enrich the fluency and clarity of the entire article. Below, we will detail the changes made and you will be able to find them all exposed and answered since the responses to each suggestion and comments are given in regular style while your comments appear in cursive. We also confirm that the writing in English has been thoroughly reviewed and accordingly improved, where needed.
- Manuscript discusses different methods of hormones removal from wastewater and evaluates their usefulness using multiple criteria analysis. In the manuscript clear information about techniques applied in the technical scale is missing.
- This note is more than accepted. The state of the art is reviewed and from line 103-109 the issue of large-scale multicriteria analysis is adequately addressed.
- The text makes impression that all mentioned techniques are still in laboratory or pilot scale. It needs completing and the recent development of the techniques of micropollutants removal as well as their current application in the technical scale should be taken into account in the presented analysis.
- This note cannot be accepted. The techniques mentioned by the reviewer are current, on a pilot and laboratory scale, but they do not have sufficient information on all the parameters considered to conduct the comparison of this scientific article. However, from lines 94-99 the state of the art on a technique implemented on a large scale is at least expanded.
- Discussed four techniques should be compared with an efficiency of an activated sludge being the most popular method of wastewater treatment. Hormones' sorption on activated sludge should also be discussed if active carbon applied directly to biological reactor as a technique is considered.
- R. The suggestion is partially accepted. The objective is focused on the comparison of the FOUR hormone abatement methods, being algae and microalgae, rotating biological discs, organic adsorbents and activated carbon. However, in lines 228-235 the technique of using activated carbon as PAC and GAC against activated sludge was widely described.
- Page 2, line 54 – “ignorance of toxicity” – this problem should be more deeply presented.
- The comment is highly welcomed. The word ignorance is changed to unawareness in line 59 and the matter is deepened. The problem is expanded in lines 60-67.
- Lines 56-59 – this sentence presents opposite opinion to the sentence given in lines 46-49 and lines 53-54, where lack of efficient removal of hormones at WWTP is stated. WWTPs base on biological processes.
- The suggestion is accepted. The text is clarified and the investigation is deepened in lines 73-77.
- Lines 59-64 – deeper discussion of the application of mentioned techniques in the technical scale is necessary.
- The suggestion is accepted. The discussion regarding oxidation, ultrafiltration and nanofiltration techniques is expanded from lines 81-93.
- Lines 93-94 – Wastewater treatment with microalgae is also applied in the technical scale. It is popular for instance in Australia and US. Some facts about it should be added.
- The observation is not accepted. Although it is true that in Australia and the US they are popular techniques for the removal of organic matter, they are not efficient for the removal of hormones, being explicitly the subject of the present investigation. In lines 132-139, more authors were referenced that corroborate what was previously mentioned.
- Figure 2 is not informative. It doesn’t present the RCB construction. If a reader doesn’t know this technology, this figure doesn’t help him to understand it. If better version of the RCB is not available it would be better to remove Figure 2 at all.
- The suggestion is obviously accepted. We removed figure 2 and did rename the other figures.
- Lines 116/117 – the sentence needs rewriting.
- The suggestion is accepted. It is better worded and additional references were placed. This correction is made in lines 163-167.
- In Figure 2 any load losses aren't seen.
- Figure 2 has been eliminated.
- Line 127 – what kind of analysis the authors mean?
- The comment is accepted. It is clarified what it means by KIND OF ANALYSIS, and it refers to the biodegradation of the effluent produced by the same WWTP. This clarification is indicated in lines 174.
- Lines 145/146 – What kind of both the effluents the authors mean?
- The comment is accepted. It is clarified to which effluents the word "both" refers. The word "both" is not correct, but the point samples are from four effluents and are characterized at four different points. This is clarified in lines 193-198.
Once again and with all due respect, we are very thankful for your comments and corrections, which helped to see a few unclear parts and or even faults of our side within our manuscript. With your comments we were able to smooth the text, clarify missing parts or wrong spellings, which resulted to a much better than the initial version of this current study. Thanks a lot on behalf of all authors
Reviewer 2 Report
The manuscript handles with methods of removal of hormones in wastewater, but with some question:
The article is well written overall.
For the better understanding of the readers, it was better to explain, as a review article is based only on 3 articles that are not from the authors.
On the other hand, it would be convenient to explain the reason for choosing the variables of different weights.
On the other hand, the conclusions would be totally different if different weights were chosen. So, what is the added value of this work?
Author Response
Response to Reviewer #2
Dear Reviewer,
As authors of the manuscript “Methods of Removal of Hormones in Wastewater”, we appreciated a lot your suggestions and comments on the document, as we are certain and convinced, that they have been useful to enrich the fluency and clarity of the entire article. Below, we will detail the changes made and you will be able to find them all exposed and answered since the responses to each suggestion and comments are given in regular style while your comments appear in cursive. We also confirm that the writing in English has been thoroughly reviewed and accordingly improved, where needed.
- The manuscript handles with methods of removal of hormones in wastewater
- The research team appreciates very much the reviewer's time for review and comment.
- The article is well written overall.
- Thanks a lot also for this comment
- For the better understanding of the readers, it was better to explain, as a review article is based only on 3 articles that are not from the authors.
- Not at all, the article presented is not based on three scientific articles. It is based on 96 authors who expose the benefits of the four methods proposed in the objective to be able to compare through a little exploited multivariate analysis technique such as modified Saatty's.
- On the other hand, it would be convenient to explain the reason for choosing the variables of different weights.
- The different weights assigned to the different variables are assigned according to the criteria of a panel of experts, as required by the chosen methodology.
- On the other hand, the conclusions would be totally different if different weights were chosen. So, what is the added value of this work?
- The conclusions would not change if different weights are assigned, since the trend would be the same, in fact identical. On the other hand, we work with a panel of experts as indicated in line 221. One part of Saaty's methodology is to analyze the degree of consistency of the variables, that is, it evaluates the degree of success with which the experts assigned the weights and is called l. This is demonstrated in equation 5.
Once again and with all due respect, we are very thankful for your comments and corrections, which helped to see a few unclear parts and or even faults of our side within our manuscript. With your comments we were able to smooth the text, clarify missing parts or wrong spellings, which resulted to a much better than the initial version of this current study. Thanks a lot on behalf of all authors